# Predictors and Long-Term Prognostic Significance of Acute Renal Function Change in Patients Who Underwent Surgical Aortic Valve Replacement

**DOI:** 10.3390/jcm12154952

**Published:** 2023-07-27

**Authors:** Dror B. Leviner, Ely Erez, Idit Lavi, Walid Saliba, Erez Sharoni

**Affiliations:** 1Department of Cardiothoracic Surgery, Carmel Medical Centre, Haifa 3436212, Israel; esharoni@clalit.org.il; 2The Ruth & Baruch Rappaport Faculty of Medicine, Technion-Israel Institute of Technology, Haifa 3525422, Israel; salibuss@yahoo.com; 3Faculty of Industrial Engineering and Management, Technion-Israel Institute of Technology, Haifa 3200003, Israel; elyerez@gmail.com; 4Department of Community Medicine and Epidemiology, Carmel Medical Centre Cardiovascular Centre, Haifa 3436212, Israel

**Keywords:** aortic valve replacement, renal function

## Abstract

There are few reports on short-term changes in renal function after surgical aortic valve replacement, and data are scarce regarding its impact on long-term outcomes. This is a retrospective study of patients who underwent isolated aortic valve replacement between 2009 and 2020 in four medical centers. Patients with end-stage renal disease were excluded. Renal function was assessed based on short-term changes. Multivariable regression models were used to identify predictors of improvement/deterioration. Cox proportional hazard models were used to assess survival trends. The study included 2402 patients, with a mean age of 69.3 years and a mean eGFR of 82.3 mL/min/1.73 m^2^. Short-term improvement rates were highest in stage 4 (24.4%) and stage 3 (16.8%) patients. Deterioration rates were highest in stage 1 (38.1%) and stage 2 (34.8%) patients. Deterioration in the chronic kidney disease stage was associated with a higher ten-year mortality (*p* < 0.001, HR 1.46); an improved stage trended toward improved survival (*p* = 0.14, HR 0.722). Patients with stage 3 and 4 kidney disease tended to remain stable or improve in the short term after aortic valve replacement while patients at stages 1 and 2 were at increased risk of deteriorating.

## 1. Introduction

Surgical and transcatheter aortic valve replacement (SAVR/TAVR) are both well-established treatment modalities for symptomatic aortic valve disease [1]. Chronic kidney disease (CKD) is an established risk factor for poor long-term survival following SAVR [2,3]. In some patients, hemodynamic instability, cardiopulmonary bypass use, and contrast medium use during TAVR may further exacerbate their renal disease [4,5]. Recent reports, however, have described an improvement in renal function following both SAVR and TAVR, especially among patients with compromised baseline renal function [6,7]. Most recently, a large prospective registry reported that acute kidney recovery (AKR) occurred in 1 out of 4 patients with baseline chronic kidney disease following TAVR [8]. The mechanism of this observed AKR has not been clearly defined but is believed to be induced by improved cardiac output following relief of aortic stenosis (AS) remediating kidney injury in patients with type 2 cardiorenal syndrome (CRS) [9].

While several studies in recent years have reported the rates of improvement and deterioration in kidney function following SAVR and their effects on short-term survival [7,10], their effects on long-term survival are unclear and the pre-operative factors predictive of these changes are poorly characterized. Additionally, previous studies have used different criteria to measure post-operative improvement/deterioration in renal function, and a gold-standard indicator has yet to be established. Our study’s primary objectives were to assess changes in renal function one-week post-SAVR, identify predictors of improvement and deterioration in post-SAVR renal function, and compare long-term survival trends. Our secondary objective was to compare two different measurement criteria of improvement/deterioration in renal function and assess their clinical relevance.

## 2. Materials and Methods

This retrospective cohort study was conducted using Clalit Health Services’ (CHS) multicenter registry, comprising four cardiothoracic centers in Israel. We reviewed the CHS registry for consecutive patients aged 18 and older who underwent isolated biological or mechanical SAVR between 1 January 2009 and 31 October 2020, and who had available pre-operative and 1-week post-operative creatinine measurements taken. Patients with multiple valve replacements, coronary artery bypass, thoracic aortic surgery, active endocarditis, and end-stage renal disease (ESRD) (CKD stage 5 or chronic dialysis use) were excluded from this study. Re-operations were also excluded. All SAVR cases were treated via median sternotomy with standard techniques. The option of a biological or mechanical valve was left to the operating surgeon’s discretion in accordance with the patient’s wishes. Each center had independent post-operative protocols regarding ICU stay, the initiation of anticoagulation, etc.

### 2.1. Demographics and Outcome Measures

CHS, the largest care provider in the Israeli national healthcare system with 4.9 million members (54% of Israel’s population), maintains a complete and comprehensive clinical and administrative data warehouse that stores patients’ lab results; medications prescribed and procured in the community; coded discharge summaries using the International Classification of Diseases, Ninth Revision (ICD-9); and billing information. Clinical and procedural characteristics, comorbidities, and laboratory values were extracted from the CHS clinical and administrative database. Patient demographic data and vital statuses were collected from the Israeli Central Bureau of Statistics and the Ministry of Internal Affairs, as previously reported [11].

Short-term renal function changes were assessed using the pre-operative and 1-week post-operative eGFR, which were calculated using the Chronic Kidney Disease Epidemiology Collaboration (CKD-EPI) equation formula [12]. The pre-operative eGFR was calculated using the last creatinine measurement before surgery (within the range of zero to six days pre-op), whereas the 1-week post-operative eGFR was calculated using the creatinine measurement taken closest to 1-week post-op (within the range of two to fourteen days post-op). Pre-operative CKD stages were delineated by the National Kidney Disease Foundation Outcomes Quality Initiative (NKF-K/DOQI) guidelines [13]: stage 1 (eGFR ≥ 90 mL/min/1.73 m^2^), stage 2 (eGFR 60–89 mL/min/1.73 m^2^), stage 3 (eGFR 30–59 mL/min/1.73 m^2^), stage 4 (eGFR 15–29 mL/min/1.73 m^2^), and stage 5 (eGFR < 15 mL/min/1.73 m^2^). The pre-operative CKD stage will also be referred to as the baseline CKD stage.

The primary outcomes were a 1-week post-operative renal function change as well as 10-year survival rates. Two measures were used to assess post-operative renal function change. The primary measure for a change in renal function was defined as an eGFR change, which is equivalent to a change in the CKD stage to that either above or below the pre-operative CKD stage (note: this is not meant to imply that the patient’s CKD stage had changed after surgery, but rather that the patients post-op eGFR measurement qualified as a higher or lower CKD stage based on the eGFR cutoffs). This will be referred to as a CKD stage improvement or CKD stage deterioration. The stage change was not limited to one stage above or below the pre-operative baseline. A secondary measure of renal function change was an increase or decrease of at least 10% in the eGFR compared to the pre-op baseline.

### 2.2. Statistical Analysis

Continuous data are presented as means (standard deviations). Categorical variables are presented as numbers (percentages). Comparisons of baseline characteristics between the three groups (improved, unchanged, and deteriorated CKD stage) were performed using one-way ANOVA for continuous variables and the chi-squared or Fisher’s exact test for categorical variables, as appropriate. Post hoc analysis of continuous variables was carried out using Tukey’s test.

Predictors of change in renal function were studied using binary logistic regression models. Two binary models were created for post-operative CKD stage changes: one for improvement and one for deterioration. Similarly, two models were created for a 10% change in the eGFR following surgery, for a total of four models. The covariates in all binary logistic regressions were age (in years), sex, pre-operative CKD stage, mechanical/biological valve, and comorbidities including diabetes, previous malignancy, hyperlipidemia, current smokers, atrial fibrillation, hypertension, cerebral vascular accident, peripheral vascular disease, chronic obstructive pulmonary disorder, and liver disease. Pre-operative serum lab values were also included, including pre-operative glucose, hemoglobin, and platelets. Age and serum lab values were continuous variables, whereas the rest of the covariates were categorical. Covariates were entered with entry/stay criteria of 0.05/0.1 in a forward stepwise fashion. The Box–Tidwell test was used to check for linear correlations of continuous variables. Variance inflation factors (VIF) were used to check for collinearity between the covariates, with a VIF value of 4 as an upper threshold. Adjusted odds ratios were used to assess the effect size.

Survival was first analyzed by a Kaplan–Meier estimator of the survival function. Since this Kaplan–Meier estimator was used to assess the survival curves of small subpopulations, the survival span analyzed was truncated at eight years to ensure a number of at least 10 at-risk patients in each population at the end of the chosen time span. Survival spans of patients with follow-up times shorter than eight years were right censored at the follow-up time available, while the survival spans of patients with follow-up times longer than eight were limited to eight years. The effects of changes in renal function on 10-year survival were assessed by two proportional hazards Cox models (one for each measure of renal function change) to adjust for possible confounding effects on long-term survival. The covariates included are detailed in the Appendix A.

Statistical analysis was performed using IBM statistics version 28 (SPSS). Missing values were assumed to be missing at random and are reported for the demographic and clinical data in Table 1. For all analyses, a *p*-value of <0.05 (2-tailed) was considered statistically significant.

## 3. Results

This study assessed 2402 of the 4184 patients who underwent surgical aortic valve replacement between 1 January 2009 and 31 October 2020 (Figure 1). The cohort’s average age was 69.3 (SD: 11.4) years old, and 38.1% were women. The mean serum creatinine and eGFR were 0.94 (SD: 0.37) mg/dl and 82.3 (SD 20.8) mL/min/1.73 m^2^, respectively. The percent of patients in each baseline CKD stage included 40.1% at stage 1, 43.5% at stage 2, 14.7% at stage 3, and 1.7% at stage 4 (Table 1).

The post-operative renal function assessment revealed changes equivalent to CKD stage improvement in 6.4% of patients, stability in 59.8%, and deterioration in 33.9%. We observed a 0.9% deterioration to stage 5 CKD. Patients with higher initial CKD stages showed higher rates of CKD stage improvement, with the most notable improvements observed in stage 4 patients (24.4%). In contrast, deterioration was most prevalent among lower baseline CKD stages, particularly stage 1 (38.1%) (Figure 2). A comparison of demographic characteristics revealed a higher prevalence of active smoking and liver disease among patients with improved stages (*p* < 0.001, *p* < 0.004) compared to the stable group. Patients with worsened stages were typically older, showed higher rates of biological valve implantation, and had a higher prevalence of pre-operative atrial fibrillation (all *p* < 0.001) (Table 1).

Analyzing renal function based on a 10% change in eGFR, 9.8% of patients showed a 10% increase, while 45.6% showed a 10% decrease, with both rates exceeding the comparable stage change metrics. The eGFR remained stable in 44.6% of patients (Figure 2). For more detailed results on eGFR changes, refer to the Appendix A.

Predictors of renal function improvement included higher pre-operative CKD stages, pre-operative hemoglobin levels, active smoking, and liver disease. Conversely, age and diabetes correlated inversely with improvement. Predictors of renal function deterioration included biological valve implantation, pre-surgery atrial fibrillation, peripheral vascular disease, and older age. Lower pre-operative CKD stages and pre-operative hemoglobin levels were inversely correlated with deterioration (Table 2). See Appendix A for predictors of a 10% change in eGFR.

The Kaplan–Meier survival estimates revealed a ten-year mortality rate of 32.1% (772/2402) (Figure 3). The Cox models showed that post-operative renal function deterioration correlated with a higher mortality rate compared to post-operative stability (*p* < 0.001, HR 1.46, CI 1.22–1.74). However, the survival benefit of improved renal function was not statistically significant (*p* = 0.14, HR 0.72, CI 0.47–1.12). A 10% eGFR reduction was also correlated with a higher mortality (*p* = 0.001, HR 1.35, CI 1.12–1.63), while a 10% increase demonstrated similar mortality rates to a stable eGFR (*p* = 0.305, HR 0.84, CI 0.60–1.17) (Figure 4). Details of the models’ covariates are provided in the Appendix A.

## 4. Discussion

Our main findings in this retrospective analysis of SAVR patients are as follows: (1) most patients with baseline CKD stages 3 and 4 had remained stable or improved in the short-term following SAVR; (2) lower baseline CKD stages were associated with higher rates of short-term stage deterioration; and (3) short-term stage deterioration was associated with a worse ten-year survival, while stage improvement trended toward an improved ten-year survival.

The correlation between poor pre-operative renal function and worse long-term survival post-SAVR is well-documented [2,14,15]. For patients suffering from CKD and severe AS, both TAVR and SAVR still provide a substantial reduction in long-term mortality compared to conservative management strategies [15], yet both TAVR and SAVR are performed far less frequently in these patients [15]. Our findings suggest that the caution of operating on such patients may not be warranted, as these patients’ renal function is far more likely to improve or remain stable than deteriorate during the immediate post-operative period. While the benefits of a post-operative stage improvement on long-term survival did not quite reach statistical significance, it was certainly trending that way and may reach significance as our sample size or follow-up duration increases. Previous work has also found similar trends in patients with more advanced CKD. Najjar et al. [10], one of the first groups to analyze AKR following SAVR, reported that patients with stage 3 and 4 CKD experienced high rates of improvement, though they employed laxer criteria for renal improvement that simply required that the eGFR increased 1-week post-SAVR. In a report by Lahoud et al. [7], AKR was more common than acute kidney injury (AKI) in patients with CKD stages 3 and 4 following SAVR or TAVR and was associated with better short-term in-hospital outcomes.

In contrast, our findings suggest that patients with low baseline CKD stages are substantially more likely to experience eGFR deterioration than improvement in the immediate post-operative period. Although patients with baseline CKD stage 1 according to our measurement criteria did not show an improvement in the eGFR, they experienced high rates of short-term stage deterioration following SAVR. The aforementioned Najjar paper [10] reported similar results, with patients classified as no CKD (stage 1 and 2 CKD grouped together) exhibiting a decrease in the eGFR from an average of 81.3 ± 14.2 mL/min/1.73 m^2^ to 74.1 ± 20.2 mL/min/1.73 m^2^. In our study, short-term deterioration rates were highest in stage 1 CKD (38.1%), followed by stage 2 with 34.8%. Furthermore, both in our cohort and in patients undergoing TAVR [16], short-term deterioration in the eGFR was a predictor of increased long-term mortality. These two stages comprised a vast majority of our cohort. In these patients it is vital that we both identify those at risk of deterioration and work to mitigate their risk. Unfortunately, most strategies aimed at the prevention of AKI after cardiac surgery have not been successful. Measures include pre-operative pharmacological interventions such as steroids [17], albumin [18], and statins [19], which mostly have had no effect or a modest effect on AKI. A possible exception which might prevent AKI is perioperative erythropoietin administration, which has been shown to be reno-protective in several randomized trials and a recent meta-analysis [20]. Intraoperative measures have also been explored. The use of minimally invasive extracorporeal circulation (MiECC) [21], the use of restrictive blood administration thresholds [22], and pharmacological interventions such as mannitol and furosemide [23,24] were all not effective in reducing AKI. Postoperative interventions include implementation of the KDIGO recommendations which showed some reduction of AKI but without reduction of renal replacement therapy [25]. In conclusion, the current ability to predict and prevent AKI following cardiac surgery is limited, but the prevalence of renal deterioration and its associated effect on long-term survival underscore the importance of developing such capabilities.

There is growing evidence suggesting that patients with a component of type 2 CRS are the ones who experience an improvement in renal function following both surgical and transcatheter aortic valve replacement (SAVR/TAVR). Patients with type 2 CRS suffer from progressive CKD caused by chronic cardiac dysfunction, which leads to reduced renal perfusion and increased renal venous congestion [9]. An improvement in cardiac output and a decrease in venous congestion are therefore believed to be the underlying mechanisms for improvement in renal function following SAVR/TAVR. Type 2 CRS is often diagnosed by exclusion, and indeed many of the factors associated with improved renal function in our cohort hint at an absence of alternative causes of renal disease. One such example is nondiabetic patients being associated with improved renal function, which matches the findings of previous reports [7,8,16]. Younger age and high baseline hemoglobin levels, which have also been previously reported [8,16,26], may be indicative of patients with fewer comorbidities associated with irreversible kidney disease. We are unsure why current smoking at the time of surgery as well as liver disease were associated with improved renal function. Deterioration in renal function following SAVR is usually attributed to hemodynamic fluctuations that are common with the use of cardiopulmonary bypass, and to the inflammatory changes that occur after SAVR. The predictors of short-term renal deterioration identified in our study, including older age, atrial fibrillation, anemia, and peripheral vascular disease, tend to represent a sicker patient with comorbid conditions. The use of a biological valve was also associated with post-op deterioration in renal function. Biological valves are unlikely to have a direct effect on renal function but are likely markers for frailer patients with more comorbidities who are more likely to receive these valves. A good baseline renal function, i.e., a low baseline CKD stage, is another predictor of deterioration in the CKD stage.

To the best of our knowledge, there is no consensus to date on the best timing post-operatively to measure changes in the eGFR to define AKR or AKI. Most studies measure renal function up to a week after surgery [7,16], while others measure renal function after longer time periods [10]. While eGFR measurements one-week post-op may not necessarily be indicative of long-term changes in patients’ renal functions, our findings suggest that these changes have a prognostic value for long-term mortality, at least in patients who experience deterioration. It is possible that measurements at later time points will be able to capture a correlation between renal function improvement and long-term survival.

While CKD is staged using the long-established NKF-K/DOQI guidelines, there is no clear standard for measuring short-term changes in renal function following surgery. Previous reports have used either a percent change in the eGFR [5,7,10,26] or a change in the CKD stage [5,16] as markers for change in renal function, making a cross-comparison of the results difficult. Our findings suggest that a change in the CKD stage is a more robust measure than a set 10% change, with a better correlation to long-term survival in both deteriorated and improved kidney function. We speculate the reason for this is that a stage change usually requires a change in the eGFR larger than 10%, and therefore represents a more substantial change in kidney function. In our study, as well as in previous reports [5,16], there were fewer patients with a change in the CKD stage than those with a 10% change in the eGFR, supporting this theory.

### Limitations

Our study is limited by its retrospective, observational nature. The small sample size of stage 4 CKD patients and the low overall rate of CKD improvement limits its power to detect differences between some subgroups, as well as its findings’ generalizability. Despite the use of multivariable Cox regression and binary logistic regression, some confounding factors might not have been accounted for, such as additional comorbidities, medication use, the etiology of aortic disease, echocardiographic features (such as pre-operative EF, pre-operative aortic valve gradients, and post-operative gradients), and surgical characteristics (i.e., size and make of an implanted valve). Due to a lack of some data, there was no surgical risk score (EUROSCORE or Society of Thoracic Surgeons score). In-hospital outcomes and complications are also not accounted for. Regarding the analysis of renal function, it is important to remember that the eGFR is an estimate of renal function. Additionally, renal function changes over time, and thus a single pre-operative measurement may not reflect a true baseline in all patients. Another limitation regarding renal function stems from the use of creatinine taken on days 2–14 after surgery (to increase the number of available patients). This might introduce bias to the eGFR.

## 5. Conclusions

The effects of SAVR on short-term renal function vary depending on patients’ baseline CKD stage. Patients with stage 3 and 4 CKD are likely to remain stable or improve after SAVR, while patients with stage 1 and 2 CKD are more likely to deteriorate. Short-term deterioration in renal function following SAVR is associated with increased long-term mortality, regardless of the baseline CKD stage. A short-term improvement in renal function trended toward improved outcomes but did not reach statistical significance. Our findings highlight the need for reno-protective strategies in cardiac surgery and suggest that the reliable prediction of post-SAVR renal function could improve pre-operative risk stratification in patients with advanced CKD.

## Figures and Tables

**Figure 1 jcm-12-04952-f001:**
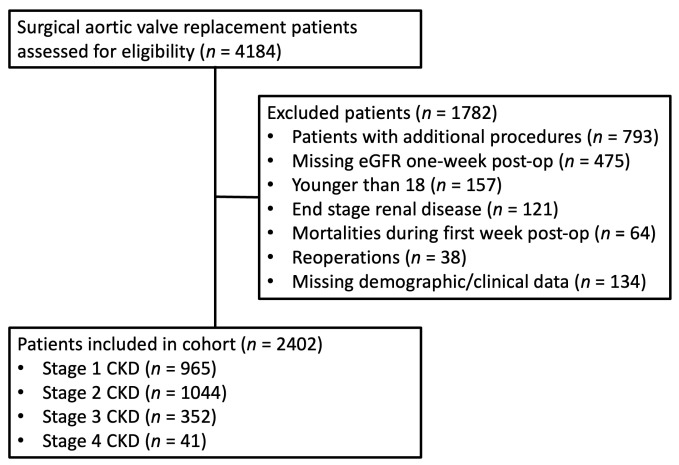
CONSORT flow chart of surgical patients undergoing aortic valve replacement included in the study. Abbreviations: CKD—chronic kidney disease; eGFR—estimated glomerular filtration rate.

**Figure 2 jcm-12-04952-f002:**
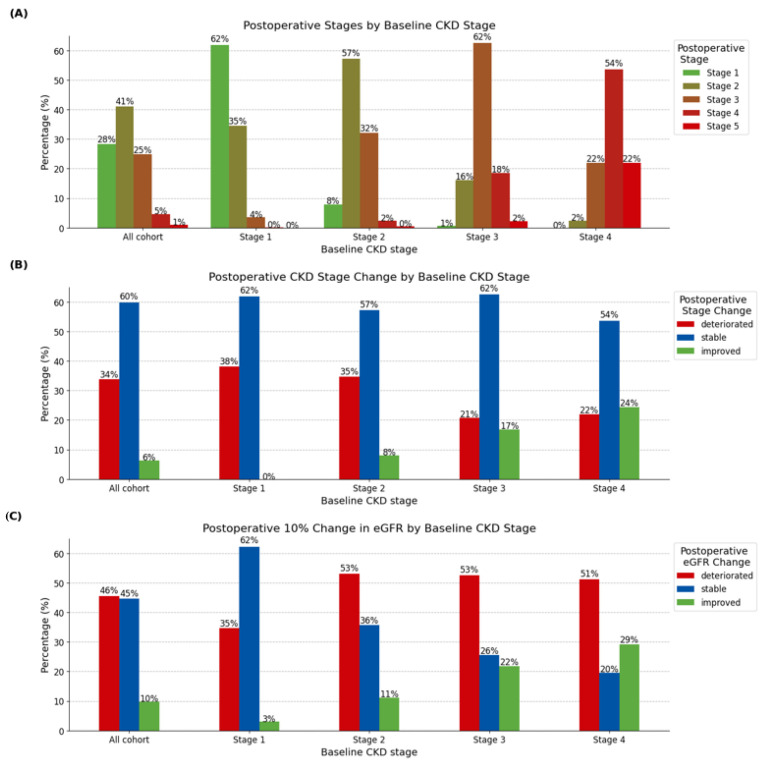
Changes in renal function one-week post-SAVR by baseline CKD stage. (**A**) Post-operative CKD stages by baseline CKD stage, (**B**) post-operative CKD stage change by baseline CKD stage, (**C**) post-operative 10% change in eGFR by baseline CKD stage. Abbreviations: CKD—chronic kidney disease; eGFR—estimated glomerular filtration rate.

**Figure 3 jcm-12-04952-f003:**
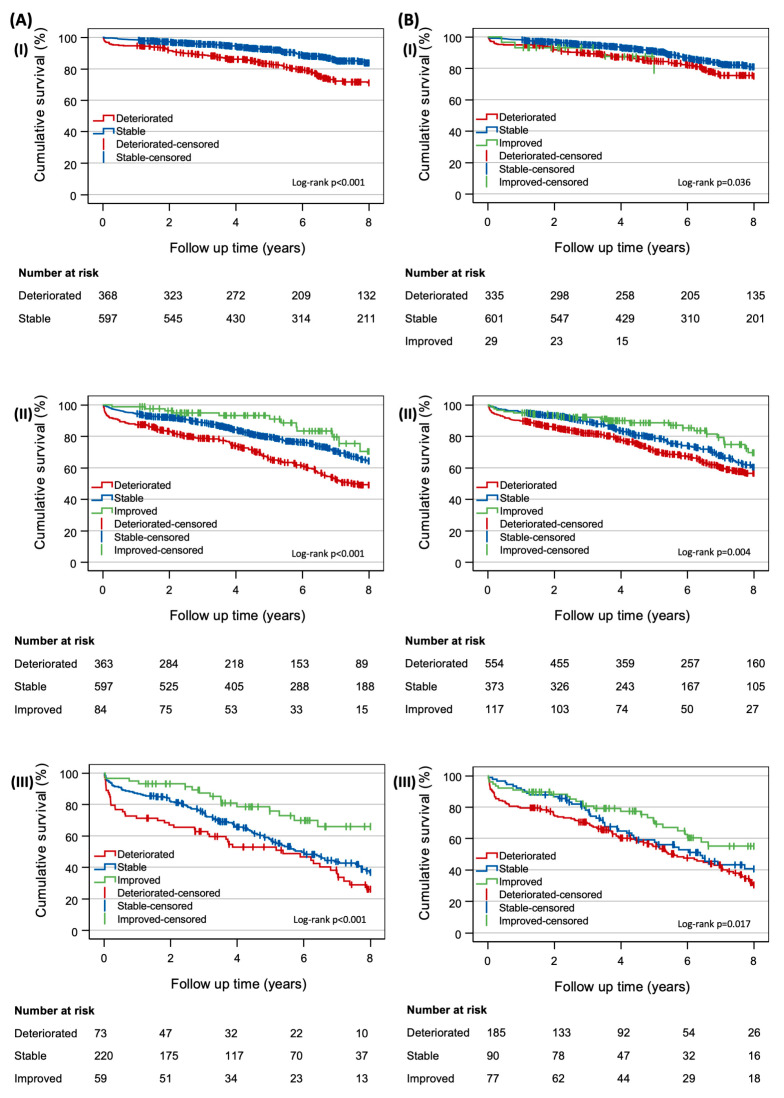
Eight-year Kaplan–Meier survival curves stratified by post-operative renal function change. (**A**) Survival curves for the following cohorts, measured by stage change: (**I**) pre-operative stage 1 CKD patients, (**II**) pre-operative stage 2 CKD patients, (**III**) pre-operative stage 3 CKD patients. (**B**) Survival curves for the following cohorts, measured by 10% change in eGFR: (**I**) pre-operative stage 1 CKD patients, (**II**) pre-operative stage 2 CKD patients, (**III**) pre-operative stage 3 CKD patients. * Baseline stage 4 CKD survival curves not shown due to small number of patients. Note: line for baseline stage 1 patients with a 10% improvement in eGFR in graph (**B**(**I**)) is truncated at the 5-year mark, due to the number at risk reaching 10 at that time. Abbreviations: CKD—chronic kidney disease; eGFR—estimated glomerular filtration rate.

**Figure 4 jcm-12-04952-f004:**
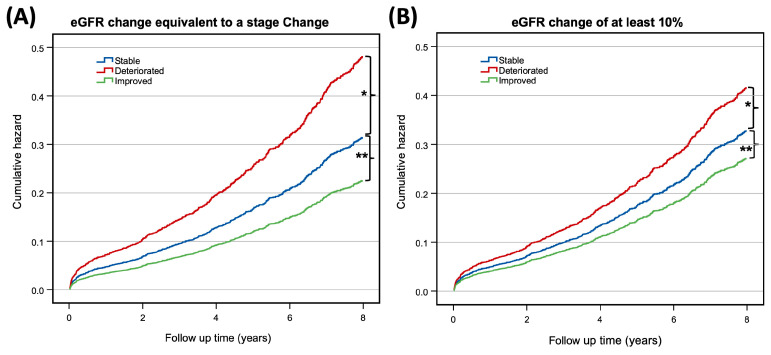
Ten-year Cox cumulative hazard function. (**A**) Cox function stratified by post-operative CKD stage change. Patients with stable, improved, and deteriorated renal function shown in blue, green, and red, respectively. (*) HR 1.46 (95% CI, 1.22–1.76), (**) HR 0.72 (95% CI, 0.47–1.12). (**B**) Cox function stratified by post-operative ±10% change in eGFR. Patients with stable, improved, and deteriorated renal function in blue, green and red, respectively. (*) HR 1.35 (95% CI, 1.12–1.63), (**) HR 0.84 (95% CI, 0.60–1.17). Abbreviations: CKD—chronic kidney disease; eGFR—estimated glomerular filtration rate; HR—hazard ratio; CI—confidence interval.

**Table 1 jcm-12-04952-t001:** Demographic and clinical characteristics by post-operative change in CKD stage.

All Cohort (*n* = 2402)	Stable Stage (*n* = 1436)	Improved Stage (*n* = 153)	Deteriorated Stage (*n* = 813)	Multiple Comp.	Stable vs. Imp.	Stable vs. Det.	Imp. vs. Det.	
	*n* (%) or Mean ± SD	*n* (%) or Mean ± SD	*n* (%) or Mean ± SD	*n* (%) or Mean ± SD	*p*-Value ^a^	Post Hoc *p*-Value	Post Hoc *p*-Value	Post Hoc *p*-Value
**Demographics**								
Age, years	69.31 (11.39)	67.94 (12.52)	66.91 (10.24)	72.18 (8.61)	<0.001 ^b^	0.532 ^d^	<0.001 ^d^	<0.001 ^d^
Women	916 (38.1)	543 (37.8)	44 (28.8)	329 (40.5)	0.022			
BMI, kg/m^2^	29.40 (5.86)	29.33 (5.76)	30.37 (8.34)	29.33 (5.45)	0.150 ^b^			
[BMI missing values]	[290]	[161]	[24]	[105]				
**Pre-op Renal Function**								
Creatinine, mg/dL	0.943 (0.365)	0.912 (0.356)	1.143 (0.406)	0.948 (0.359)	<0.001 ^b^	<0.001 ^d^	0.067 ^d^	<0.001 ^d^
eGFR, ml/min/1.73 m^2^	82.28 (20.79)	84.67 (21.70)	67.22 (18.47)	80.91 (18.10)	<0.001 ^b^	<0.001 ^d^	<0.001 ^d^	<0.001 ^d^
**Baseline CKD**					<0.001 ^c^			
Stage 1	965 (40.2)	597 (41.6)	0	368 (45.3)				
Stage 2	1044 (43.5)	597 (41.6)	84 (54.9)	363 (44.6)				
Stage 3	352 (14.7)	220 (15.3)	59 (38.6)	73 (9.0)				
Stage 4	41 (1.7)	22 (1.5)	10 (6.5)	9 (1.1)				
**Valve type**					<0.001			
Biological	1487 (61.9)	838 (58.4)	91 (59.5)	558 (68.6)				
Mechanical	915 (38.1)	598 (41.6)	62 (40.5)	255 (31.4)				
**Comorbidity**								
Smoking	1099 (45.8)	662 (46.1)	87 (56.9)	350 (43.1)	0.007			
CAD	245 (10.2)	136 (9.5)	15 (9.8)	94 (11.6)	0.286			
AF	769 (32.0)	419 (29.2)	44 (28.8)	306 (37.6)	<0.001			
CVA	321 (13.4)	177 (12.3)	22 (14.4)	122 (15.0)	0.186			
COPD	263 (10.9)	148 (10.3)	21 (13.7)	94 (11.6)	0.345			
Diabetes	997 (41.5)	589 (41.0)	57 (37.3)	351 (43.2)	0.331			
Liver disease	53 (2.2)	27 (1.9)	10 (6.5)	16 (2.0)	0.004^c^			
Cancer	154 (6.4)	95 (6.6)	6 (3.9)	53 (6.5)	0.438			
Hyperlipidemia	209 (8.7)	125 (9.2)	16 (10.5)	68 (8.4)	0.686			
Hypertension	1764 (73.4)	1030 (71.7)	112 (73.2)	622 (76.5)	0.045			
PVD	247 (10.3)	127 (8.8)	17 (11.1)	103 (12.7)	0.015			
**Pre-operative serum values**								
Hgb (g/dL)	11.47 (1.35)	11.57 (1.34)	11.69 (1.59)	11.25 (1.29)	<0.001 ^b^	0.503 ^d^	<0.001 ^d^	0.001 ^d^
[missing]	[4]	[3]	[0]	[1]				
Glucose (mg/dL)	148.92 (39.87)	147.30 (39.02)	151.50 (38.77)	151.30 (41.44)	0.052 ^b^			
Platelets (10^3^/µL)	192.06 (61.96)	192.80 (61.91)	202.47 (58.37)	188.81 (62.51)	0.035 ^b^	0.161 ^d^	0.307 ^d^	0.034 ^d^
[missing]	[5]	[3]	[1]	[1]				

^a^ Pearson chi-squared test, unless otherwise stated; ^b^ one way ANOVA; ^c^ Fisher’s exact test; ^d^ Tukey’s test. Abbreviations: AF—atrial fibrillation; BMI—body mass index; CAD—coronary artery disease; CKD—chronic kidney disease; comp—comparison; COPD—chronic obstructive pulmonary disease; CVA—cerebrovascular accident; det—deteriorated; eGFR—estimated glomerular filtration rate; imp—improved; PVD—peripheral vascular disease.

**Table 2 jcm-12-04952-t002:** Binary logistic regression results for renal improvement and renal deterioration by CKD stage change.

Predictors of Renal Improvement	Adjusted Odds Ratios (95% CI)	*p*-Value	Predictors of Renal Deterioration	Adjusted Odds Ratios (95% CI)	*p*-Value
**Preoperative stage 3 CKD ^a^**	**3.013 (1.973, 4.600)**	**<0.001**	**Preoperative stage 2 CKD ^b^**	**0.540 (0.433, 0.672)**	**<0.001**
**Preoperative stage 4 CKD ^a^**	**5.243 (2.081, 13.211)**	**<0.001**	**Preoperative stage 3 CKD ^b^**	**0.263 (0.189, 0.366)**	**<0.001**
**Biological valve**	0.976 (0.656, 1.453)	0.905	**Preoperative stage 4 CKD ^b^**	**0.295 (0.130, 0.671)**	**0.004**
**Diabetes**	**0.515 (0.330, 0.805)**	**0.004**	**Biological valve**	**0.720 (0.595, 0.873)**	**<0.001**
**Previous malignancy**	0.456 (0.181, 1.144)	0.094	Diabetes	0.943 (0.763, 1.167)	0.591
**Hyperlipidemia**	1.004 (0.525, 1.919)	0.991	Previous malignancy	0.939 (0.653, 1.351)	0.736
**Active smoking**	**1.736 (1.123, 2.682)**	**0.013**	Hyperlipidemia	1.131 (0.815, 1.568)	0.462
**Atrial fibrillation**	0.892 (0.585, 1.359)	0.594	Current smoker	0.902 (0.736, 1.106)	0.323
**Hypertension**	0.752 (0.454, 1.245)	0.268	**Atrial fibrillation**	**1.350 (1.112, 1.638)**	**0.002**
**Cerebral vascular accident**	1.047 (0.604, 1.812)	0.871	Hypertension	0.973 (0.771, 1.228)	0.818
**Peripheral vascular disease**	0.863 (0.465, 1.602)	0.640	Cerebral vascular accident	1.071 (0.820, 1.398)	0.617
**COPD**	1.171 (0.654, 2.096)	0.596	**Peripheral vascular disease**	**1.437 (1.067, 1.935)**	**0.017**
**Liver disease**	**3.079 (1.211, 7.826)**	**0.018**	COPD	1.072 (0.798, 1.440)	0.643
**Female**	1.059 (0.68, 1.650)	0.798	Liver disease	1.009 (0.526, 1.934)	0.979
**Age (years)**	**0.933 (0.914, 0.952)**	**<0.001**	Female	0.890 (0.727, 1.090)	0.260
**Preoperative glucose (mg/dL)**	1.003 (0.998, 1.007)	0.284	**Age (years)**	**1.048 (1.037, 1.060)**	**<0.001**
**Preoperative hemoglobin (g/dL)**	**1.225 (1.068, 1.405)**	**0.004**	Preoperative glucose (mg/dL)	1.002 (0.999, 1.004)	0.139
**Preoperative platelets (10^3^/µL)**	1.002 (0.999, 1.005)	0.104	**Preoperative hemoglobin (g/dL)**	**0.842 (0.781, 0.907)**	**<0.001**
			Preoperative platelets (10^3^/µL)	0.999 (0.998, (1.001)	0.296

^a^ Compared to reference stage 2 CKD; ^b^ compared to reference stage 2 CKD. Abbreviations: CKD—chronic kidney disease; COPD—chronic obstructive pulmonary disease.

## Data Availability

Data will be made available upon reasonable request.

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
