# Peer review of "Predictors and Long-Term Prognostic Significance of Acute Renal Function Change in Patients Who Underwent Surgical Aortic Valve Replacement"

_jcm, 2023, doi:10.3390/jcm12154952_

Round 1

Reviewer 1 Report

I am honored to have the opportunity to review this paper.

As noted in the limitation, I frankly feel that the lack of consideration of hemodynamic data and drug use in discussing renal function after AVR and its prognosis is inadequate as a design. On the other hand, the number of cases is relatively large and the results are acceptable.

If I may make one comment, I think Table 2 is a little difficult to read and should be a Figure.

Author Response

I am honored to have the opportunity to review this paper.

As noted in the limitation, I frankly feel that the lack of consideration of hemodynamic data and drug use in discussing renal function after AVR and its prognosis is inadequate as a design. On the other hand, the number of cases is relatively large and the results are acceptable.

If I may make one comment, I think Table 2 is a little difficult to read and should be a Figure.

Answer: Thank you for this comment. Table 2 has now been changed to Figure 2

Reviewer 2 Report

.Review for

Predictors and long-term prognostic significance of acute renal 2 function change in patients who underwent surgical aortic 3 valve replacement

Interesting retrospective analysis to determine renal function change in SAVR/TAVR patients. 

Maybe a longer observational period and blood sample analysis would have been better for understanding the mechanism mostly in Stage 3,4. 

Moreover excluding stage 5 patients or chronically dialyzed patients' data is potentially not a good idea. 

I would have been interesting to assess preoperative risk stratification data as written in the Limitations section. Besides to examine two population biological vs mechanical valve would be the most interesting since indication (not only age) is different.

The potential of AVR+ACBG or AVR+MVR/MVP and ACBG is not mentioned. It would have been interesting to see data in these subgroups.

10% change in GFR is measured. Would it make sense to mine data and see 20% change foer example?

Table 1, 2 should be more conscious.

Results section should be a bit more conscious, should be rewritten.

It would be elegant to have more data, reference on potential kidney injury prevention.

In line 52 renal ... and assess. I suppose function is missing.

Author Response

Thank you for a thoughtful review. We did our best to answer your comments and improve the article accordingly. Please see our responses bellow 

Maybe a longer observational period and blood sample analysis would have been better for understanding the mechanism mostly in Stage 3,4.

Answer: Thank you for this comment. There were two reasons we did our analysis with short term changes. The first is we wanted to know if we would find a connection between short time changes and long term follow up. This would give the surgeon a picture of the possible outcome of the patient during the initial hospitalization. The second reason was that we did not want to introduce bias into the study. Since the data was not collected prospectively, patients with deterioration in renal function (or those with overzealous primary care physicians for that matter) would have renal function tests months after surgery whereas other patients might not. This is definitely an interesting prospect for future studies.

Moreover excluding stage 5 patients or chronically dialyzed patients' data is potentially not a good idea.

Answer: Since stage 5 patients cannot improve or deteriorate, and this was our primary outcome, we chose to exclude them from the present study. We hope this answers your comment

I would have been interesting to assess preoperative risk stratification data as written in the Limitations section. Besides to examine two population biological vs mechanical valve would be the most interesting since indication (not only age) is different.

Answer: Thank you for your comment. Unfortunately, we had difficulty getting granular data to allow us to calculate risk scores. As for valve type, we looked at the effects of valve type on both renal function change and survival, examining the differences solely on valve type is an Interesting perspective for future studies. 

The potential of AVR+ACBG or AVR+MVR/MVP and ACBG is not mentioned. It would have been interesting to see data in these subgroups. 

Answer: We wanted to focus our first study on this subject on as “clean” as possible population. That is the reason we focused only on isolated AVR patients. Mixed valve patients and those with valve + ACBG are a more heterogenous group with more common risk factors for kidney disease apart from aortic valve disease. We will definitely consider repeating this study with a larger and more heterogenous patient population.

10% change in GFR is measured. Would it make sense to mine data and see 20% change foer example?

Answer: We used a 10% change based on previous published studies which used the same cutoff in order to try and maintain consistency with previous published studies (reference 5 for example - Witberg G, Steinmetz T, Landes U, et al. Change in Kidney Function and 2-Year Mortality After Transcatheter Aortic Valve Replacement. JAMA Netw Open. 2021;4(3):e213296. doi:10.1001/jamanetworkopen.2021.3296)

Table 1, 2 should be more conscious.

Answer: Thank you for your comment. We did our best to shorten table 1. As for table 2, in accordance with the request of another reviewer, we made a figure based on it (figure 2) and deleted the table altogether.

Results section should be a bit more conscious, should be rewritten.

Answer: The results section has been completely rewritten to make it more concise.

 It would be elegant to have more data, reference on potential kidney injury prevention.

Answer: Thank you for this comment. We elaborated a bit more in the discussion on potential kidney injury prevention strategies (unfortunately mostly are unsuccessful). Please see page 15, lines 379-386

Reviewer 3 Report

This paper evaluates the effect of changes in renal function in patients after a valve replacement. Predictably, those whose renal function deteriorated were more likely to die. The claim in the abstract that if there was an improvement they were less likely to die formed a 'trend' because P was not significant is laughable and should be removed.

Methods:

Whilst the 1-week post-op time range is stated to be 2days - 2 weeks, the pre-op time range is not stated. Is that because the 1week pre-op sample could have been taken a very long time before the operation and would therefore be technically invalid / embarrassing because it shows poor pre-op testing? The time window, mean, range etc., needs to be given for pre-op as well as for post-op.

Overall however, a well-presented paper with some interesting findings,

Author Response

This paper evaluates the effect of changes in renal function in patients after a valve replacement. Predictably, those whose renal function deteriorated were more likely to die. The claim in the abstract that if there was an improvement they were less likely to die formed a 'trend' because P was not significant is laughable and should be removed.

Answer: Thank you for this comment. We removed this sentence from the abstract.

Methods:

Whilst the 1-week post-op time range is stated to be 2days - 2 weeks, the pre-op time range is not stated. Is that because the 1week pre-op sample could have been taken a very long time before the operation and would therefore be technically invalid / embarrassing because it shows poor pre-op testing? The time window, mean, range etc., needs to be given for pre-op as well as for post-op.

Answer: Thank you for this comment. The time frame was up to six days before surgery. We added this to the methods section. Page 2, line 81 “(within the range of zero to six days pre-op)”

Overall however, a well-presented paper with some interesting findings,